

# A comprehensive analysis of the oncogenic and prognostic role of TBC1Ds in human hepatocellular carcinoma

Pei Zhang*, Lei Zhu* and Xiaodong Pan

The Affiliated Geriatric Hospital of Nanjing Medical University, Nanjing, China
* These authors contributed equally to this work.

## ABSTRACT

**Backgrounds:** TBC1D family members (TBC1Ds) are a group of proteins that contain the Tre2-Bub2-Cdc16 (TBC) domain. Recent studies have shown that TBC1Ds are involved in tumor growth, but no analysis has been done of expression patterns and prognostic values of TBC1Ds in hepatocellular carcinoma (HCC).
**Methods:** The expression levels of TBC1Ds were evaluated in HCC using the TIMER, UALCN and Protein Atlas databases. The correlation between the mRNA levels of TBC1Ds and the prognosis of patients with HCC in the GEPIA database was then analyzed. An enrichment analysis then revealed genes that potentially interact with TBC1Ds. The correlation between levels of TBC1Ds and tumor-infiltrating immune cells (TIICs) in HCC were studied using the TIMER 2.0 database. Finally, a series of *in vitro* assays verified the role of TBC1Ds in HCC progression.
**Results:** This study revealed the upregulated expression of TBC1Ds in HCC and the strong positive correlation between the mRNA levels of TBC1Ds and poor prognosis of patients with HCC. The functions of TBC1Ds were mainly related to autophagy and the AMPK pathway. There was also a significant correlation between level of TBC1Ds and tumor-infiltrating immune cells (TIICs) in HCC. The promoting role of TBC1Ds in HCC progression was verified *in vitro* assays.
**Conclusion:** The results of this analysis indicate that TBC1Ds may serve as new biomarkers for early diagnosis and treatment of HCC.

## INTRODUCTION

Hepatocellular carcinoma (HCC) remains the fourth most common cancer worldwide (*Siegel, Miller & Jemal, 2019*). Although numerous advanced achievements have been made in chemical therapies in recent years, surgery is still the first-line treatment for HCC. Owing to the heterogeneity and complexities of HCC, it is difficult to diagnose early, so the survival rate for patients with HCC is only 40% because of intrahepatic metastasis and a high postoperative recurrence rate (*Papatheodoridis et al., 2020*). Therefore, it is important to discover the specific molecular mechanisms underlying the occurrence and progression of HCC.

TBC1D family members (TBC1Ds) are a group of proteins that contain the Tre2-Bub2-Cdc16 (TBC) domain. TBC1Ds are unique in that they use a dual finger mechanism to

Corresponding author
Xiaodong Pan,
panxiaodong120@outlook.com

inactivate their targets (*Duan et al., 2021*). In recent years, autophagy and glucose metabolism have been shown to have vital roles in the pathogenesis and metastasis of HCC. In the early cancer stages, especially during the tumor pathogenesis process, the disturbance of autophagy can often be a partial carcinogenic factor. However, autophagy can also promote the survival and chemo-resistance of cancer cells in the development process (*Amaravadi, Kimmelman & Debnath, 2019*). In most types of cancers, the metabolism shift of relying on lactic acid fermentation even in oxygen rich conditions, termed aerobic glycolysis, is the hallmark of carcinogenesis (*Reinfeld et al., 2022*). Many studies have shown the close association between autophagy and glucose metabolism (*Zhang et al., 2021*). Glucose deprivation can induce autophagy, primarily through AMPK activation; in return, autophagy helps improve the glucose metabolism of cancer cells (*Yang & Klionsky, 2021*).

This study focused on several TBC1Ds, which have been shown to participate in the processes of autophagy and glucose metabolism. TBC1D1 participates in the process of glucose transport and can influence the outcome of nonalcoholic fatty liver disease (*De Wendt et al., 2021*; *Chen et al., 2022*). TBC1D7 is the third functional subunit of the TSC1-TSC2 complex upstream of MTORC1, which plays an important role in autophagy. Loss of TBC1D7 function can lead to increased activation of MTORC1 signaling, delayed induction of autophagy and upregulated cell growth under poor growth conditions (*Dibble et al., 2012*). TBC1D8 interacts with PKM2 *via* its Rab-GAP TBC domain to promote aerobic glycolysis in ovarian cancer (*Chen et al., 2019*). Previous studies have also found that TBC1D9B can colocalize with LC3 in autophagosomes and interact with several mammalian ATG8 homologs. When TBC1D9B is inhibited, cellular autophagy activity is reduced, suggesting TBC1D9B is involved in autophagic flux (*Liao et al., 2018*; *Gallo et al., 2014*). TBC1D14 has also been reported to colocalize and interact with ULK1, which is involved in the early formation of autophagosomes. When overexpressed, TBC1D14 causes tubulation of ULK1-positive endosomes and inhibits autophagosome formation. TBC1D14 relocates to the Golgi complex when cells are in amino acid starvation (*Longatti et al., 2012*). TBC1D25 can be recruited to the outer membrane of phagophores and autophagosomes by interacting with ATG8 family proteins (*Itoh et al., 2008*). Since the pathogenesis and diagnosis of HCC are complex, this study attempted to reveal a new perspective of HCC and explore the possibility of TBC1Ds serving as new molecular markers for HCC.

## MATERIALS AND METHODS

### TIMER database analysis

The Tumor Immune Estimation Resource (TIMER) is a comprehensive database that provides systematic analysis of gene expression and immune cells infiltrates in 32 different types of cancers (https://cistrome.shinyapps.io/timer/) from The Cancer Genome Atlas (TCGA). TBC1Ds were input into the TIMER database and the expression difference of TBC1Ds were observed between tumor and adjacent normal tissues for the different tumors or specific tumor subtypes in the TCGA. The "Immune-Gene" module of the

TIMER database was used to verify the association between TBC1D expression and tumor-infiltrating immune cells, including B cells, CD4+ T cells, CD8+ T cells, neutrophils, macrophages and dendritic cells (DCs), in HCC.

### UALCAN database analysis

UALCAN (http://ualcan.path.uab.edu/) is an interactive web resource, based on TCGA, for analyzing the relative gene expression in tumor tissues and normal tissues. In this study, UALCAN was used to analyze the relative mRNA expression of TBC1Ds across tumor grades. The promoter methylation levels of TBC1Ds in HCC were also analyzed with this tool.

### GEPIA database analysis

Gene Expression Profiling Interactive Analysis (GEPIA; http://gepia.cancer-pku.cn/) is a newly developed interactive database for gene expression analysis of tissue samples from the TGCA and GTEx projects. This study used GEPIA to analyze the relative gene expression of TBC1Ds in HCC tumor tissues and paratumor tissues. The correlation between TBC1Ds and survival in patients with HCC was also explored using GEPIA. The correlation analysis of TBC1Ds and AFP, PDL1 and CTLA4 were also performed with the GEPIA database.

### cBioPortal database analysis

The cBioPortal database (http://www.cbioportal.org/) is an open-access web tool for analyzing multidimensional cancer genomics datasets. In this study, the multidimensional analysis included mutation, copy number variation (CNV), and association of TBC1D mutations with clinical features in HCC.

### STRING and metascape database analysis

The STRING (search tool for the retrieval of interacting genes/proteins) is a database of interactions between genes or proteins (https://cn.string-db.org/). In this study, STRING was used to predict possible protein-protein interaction networks between TBC1Ds and associated genes. The Metascape database (https://metascape.org/gp/index.html#/main/step1) is a web-based tool that combines gene, interactome, and functional enrichment analyses. This study used the Metascape database to perform enrichment analyses of the functional roles of TBC1Ds and their predicted associated genes.

### Human protein atlas database analysis

The Human Protein Atlas is a Swedish-based program that maps all the human proteins in cells, tissues and organs using an integration of various omics technologies, including antibody-based imaging, mass spectrometry-based proteomics, transcriptomics and systems biology (https://www.proteinatlas.org/). This study used the Human Protein Atlas to explore the protein levels of TBC1Ds in tumor and paratumor tissues of patients with HCC.

## Kaplan–Meier plotter database analysis

The Kaplan–Meier plotter database (http://kmplot.com/analysis/) provides data analysis of 10,461 clinical samples of patients, including gastric cancer, breast cancer and HCC samples. In this study, the Kaplan–Meier plotter database was used to examine the association between TBC1Ds and the relapse free survival (RFS) of patients with HCC.

## Immunohistochemistry

The HCC sections were dewaxed at 60 °C for an hour, then soaked through with a dimethylbenzene and ethanol solution. Sections were then incubated with rabbit polyclonal antibody against TBC1D14 (all from Abclonal) at 1/200 dilution overnight at 4 °C and antibody conjugated with peroxidase (HRP; 1:500 dilution) at room temperature for 2 h, then covered by 3,3-diaminobenzidine (DAB). HCC histologic sections stained for TBC1D14 were purchased from Servicebio.

## Western blotting

Proteins were extracted from cells with western blotting kit according to manufacturer's instructions (Beyotime, Shanghai, China). Then the protein solutions were separated on 10% SDS-PAGE gels, and transferred to a PVDF membrane (Millipore). After that, the membranes were blocked for 2 h at room temperature and incubated with TBC1D8 (#sc-376637, 1:1,000; Santa Cruz Biotechnology, Dallas, TX, USA) or TBC1D14 (#ab235544, 1:1,000; Abcam, Cambridge, UK) antibodies overnight at 4 °C. The membranes were visualized with an ECL detection system after incubation with secondary antibody at room temperature for 2 h.

## Cell culture and transfection

Human HCC cell line (Hep3B) was purchased from the Shanghai Institute of Biochemistry and Cell Biology and cultured in Dulbecco's modified Eagle's medium (DMEM) with 10% fetal bovine serum (FBS) and 1% penicillin-streptomycin (WISENT, Shanghai, China). Before the cells were harvested, they were cultured at 37 °C with 5% CO2. The Cas9 expression construct pLV1-CMV-Cas9-Puro-U6-sgRNA was used for the expression of sgRNAs in Hep3B cells; sgRNA targeting TBC1D8 was 5′-AAATGGAGCGACCCGTGCAT-3′, and sgRNA targeting TBC1D14 was 5′-GGAATCCCTCCAAGTGTGAG-3′. Empty pLV1-CMV-Cas9-Puro-U6-sgRNA plasmid was used as a control (sgNC). Puromycin (1 µg/ml, Merck, 540,411) was added 24 h after transfection. Cells were collected 72 h after transfection for subsequent experiments.

## Cell proliferation and migration assays

The EdU Assay Kit (Beyotime, Shanghai, China) and CCK8 Kit (Vazyme, Nanjing, China) were used to examine the growth ability of HCC cells. For the EdU assay, Hep3B cells were seeded into six-well plates at a $4 * 10^5$ cells/well density and incubated with EdU according to the manufacturer's instructions. For the CCK8 assay, Hep3B cells were seeded into 96-well plates at a density of 1,000 cells/well, and five multiple wells were designed for each

group. After incubation for 24, 48 and 72 h, the CCK8 solution was added and the cell viability were measured according to manufacturer's instructions.

For the wound healing assay, Hep3B cells were seeded into six-well plates at a density of $4 * 10^5$ cells/well with serum-free medium. Then, the cells were scratched with 200 µl pipette tips. Cell migration was observed by photography in an Olympus CKX41 inverted microscope at 0 and 48 h.

## RESULTS

### Transcriptional level of TBC1Ds in patients with HCC

Six TBC1D factors were found to have vital functions in autophagy. The TIMER 2.0 database was used to investigate the transcriptional levels of TBC1Ds in cancers. Compared with the mRNA of TBC1Ds in normal samples, the mRNA of TBC1Ds were strongly upregulated in a variety of cancers including colon cancer, cholangiocarcinoma and pancreatic cancer. The most important finding was that the mRNA levels of all six TBC1D members–TBC1D1, TBC1D7, TBC1D8, TBC1D9b, TBC1D14 and TBC1D25— were significantly upregulated in HCC tissues and cholangio carcinoma (CHOL) tissues (Fig. 1A and Fig. S1). The Gene Expression Omnibus (GSE102079) database also confirmed the upregulated mRNA levels of TBC1Ds in HCC tissues compared to paratumor tissues (Fig. S2). To further verify the results, the mRNA levels of TBC1Ds were also explored in the UALCN dataset and the expression levels of TBC1D1, TBC1D7, TBC1D8, TBC1D9b, TBC1D14 and TBC1D25 were significantly higher in HCC than in normal liver tissues (Fig. 1B). These results all indicated that TBC1Ds are abnormally upregulated in HCC.

### Verified protein levels of TBC1Ds in human HCC tissues

The Human Protein Atlas database and immunohistochemistry technology were employed to demonstrate the protein levels of TBC1Ds in HCC. As shown in the graph, TBC1D1, TBC1D8, TBC1D9b, TBC1D14 and TBC1D25 were highly expressed in HCC tissues but moderately expressed in normal liver tissues. However, there were no significant differences in TBC1D7 expression between HCC and normal tissues (Fig. 2). The IHC value was also analyzed and the results are shown in Fig. S3. These results further verified the abnormal upregulation of TBC1Ds in HCC.

### Relationship between the expression of TBC1Ds and the clinicopathological parameters of patients with HCC

Using the GEPIA dataset, the mRNA levels of TBC1Ds were also analyzed at different cancer stages of HCC. The results showed that the mRNA levels of TBC1D1, TBC1D7, TBC1D8 and TBC1D14 significantly varied between different cancer stages, whereas the mRNA levels of TBC1D9b and TBC1D25 did not (Fig. 3A). The mRNA transcription levels of TBC1Ds were also found to be significantly upregulated as the tumor grades increased (Fig. 3B). These results indicated that TBC1Ds are highly expressed in HCC and may serve as vital biomarkers for HCC diagnosis.

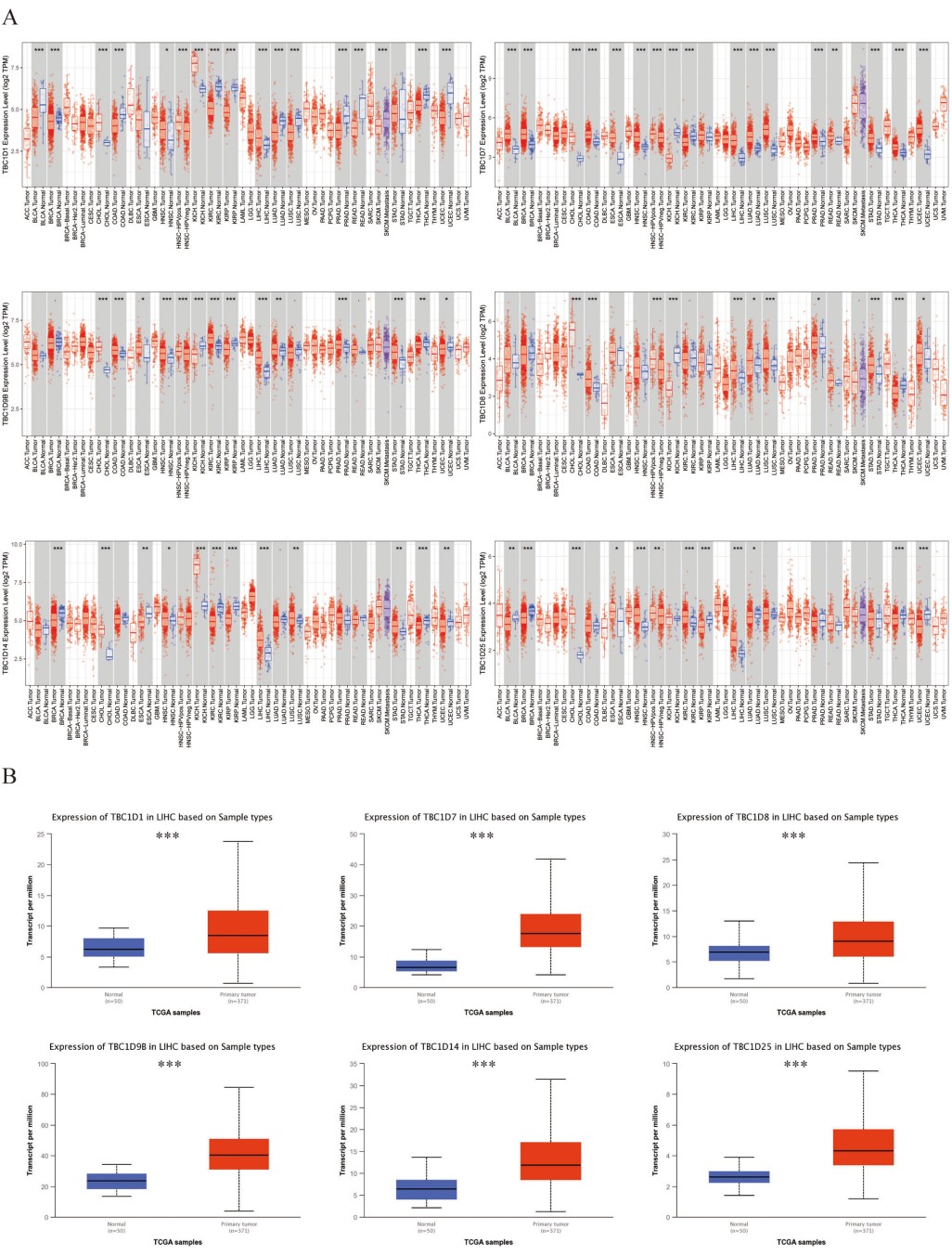

**Figure 1 Transcript level of TBC1Ds in tumor and paratumor tissues.** (A) Expression of TBC1Ds in different tumor types from the TCGA database in TIMER. (ACC, BLCA, CESC, CHOL, COAD, DLBC, ESCA, GBM, HNSC, KICH, KIRC, KIRP, LAML, LGG, LIHC, LUAD, LUSC, MESO, OV, PAAD, PCPG, PRAD, READ, SARC, SKCM, SKCM, TGCT, HCA, THYM, UCEC, UCS, UVM). (B) The box plot show the mRNA expression of TBC1Ds in tumor tissues and adjacent healthy tissues, according to the t-test in UALCN. (*$P < 0.05$, **$P < 0.01$, ***$P < 0.001$).

The alpha fetoprotein (AFP) is one of the most widely used biomarkers for the clinical diagnosis of HCC, and levels of serum AFP are closely correlated with tumor development (*Heimbach et al., 2018*; *Wong, Ahmed & Gish, 2015*; *Li et al., 2009*). The mRNA levels of

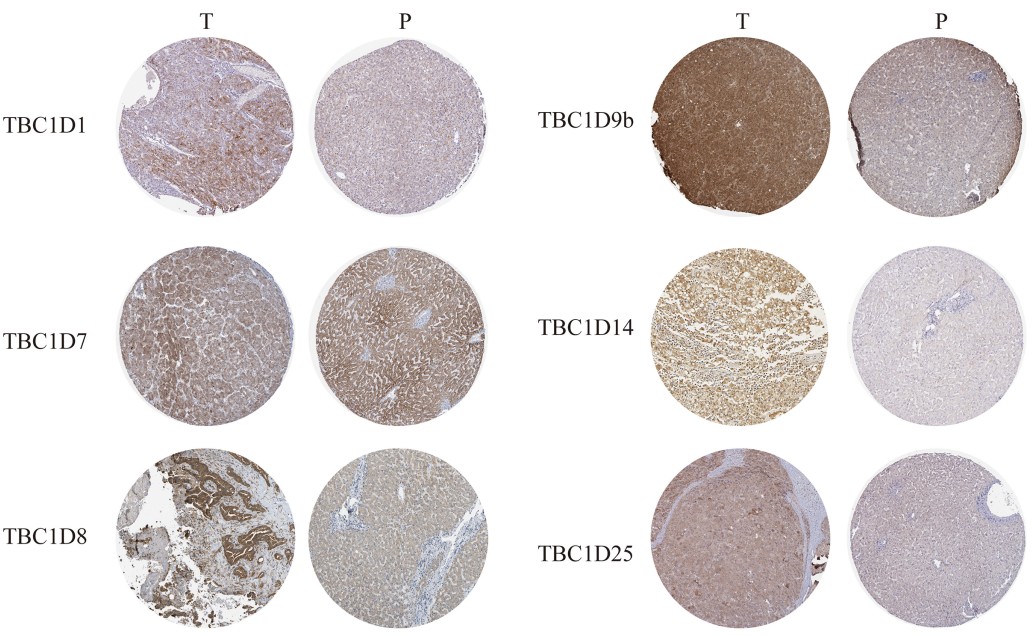

**Figure 2 The protein expressions of TBC1Ds in HCC.** Representative immunohistochemistry analysis of TBC1D1, TBC1D7, TBC1D8, TBC1D9b and TBC1D25 from the human protein atlas database. And we performed immunohistochemistry experiment to verify TBC1D14 protein expression in HCC.

TBC1D1, TBC1D7 and TBC1D9b were significantly positively correlated with the mRNA level of AFP (Fig. 3C) in the GEPIA database. However, the association between TBC1Ds and AFP still needs to be validated in future experiments.

## TBC1D expression was positively correlated with patient outcome

This study also explored the critical efficiency of TBC1Ds in the survival of patients with HCC. The GEPIA tool showed that high expressions of TBC1D1, TBC1D7 and TBC1D9B were significantly correlated with poor overall survival (OS) of patients with HCC. A disease-free survival analysis data (DFS) showed a significant correlation between high TBC1D8 and TBC1D14 expression and poor prognosis of patients with HCC (Fig. 4). However, only the high transcriptional level of TBC1D8 was significantly correlated with the relapse free survival (RFS) of patients with HCC (Fig. S4). These results suggested the prognostic significance of TBC1D expression and its close relationship with the clinical characteristics of HCC.

## Genomic alterations of TBC1Ds in HCC

To investigate the mutations of TBC1D genes in HCC, the cBioPortal database was used to analyze the DNA sequencing data from patients with HCC. The TBC1Ds were altered in 305 (88%) samples of the 348 total HCC patient samples (Fig. 5A). Notably, single mutations of both TBC1D1 and TBC1D7 were found in almost half of the patients with HCC (Fig. 5B). The alterations involved mRNA missense mutation, splice mutation, truncating mutation, amplification, deep deletion, high mRNA and low mRNA. The most common alteration of TBC1D1, TBC1D8, TBC1D14 and TBC1D25 was shallow deletion,

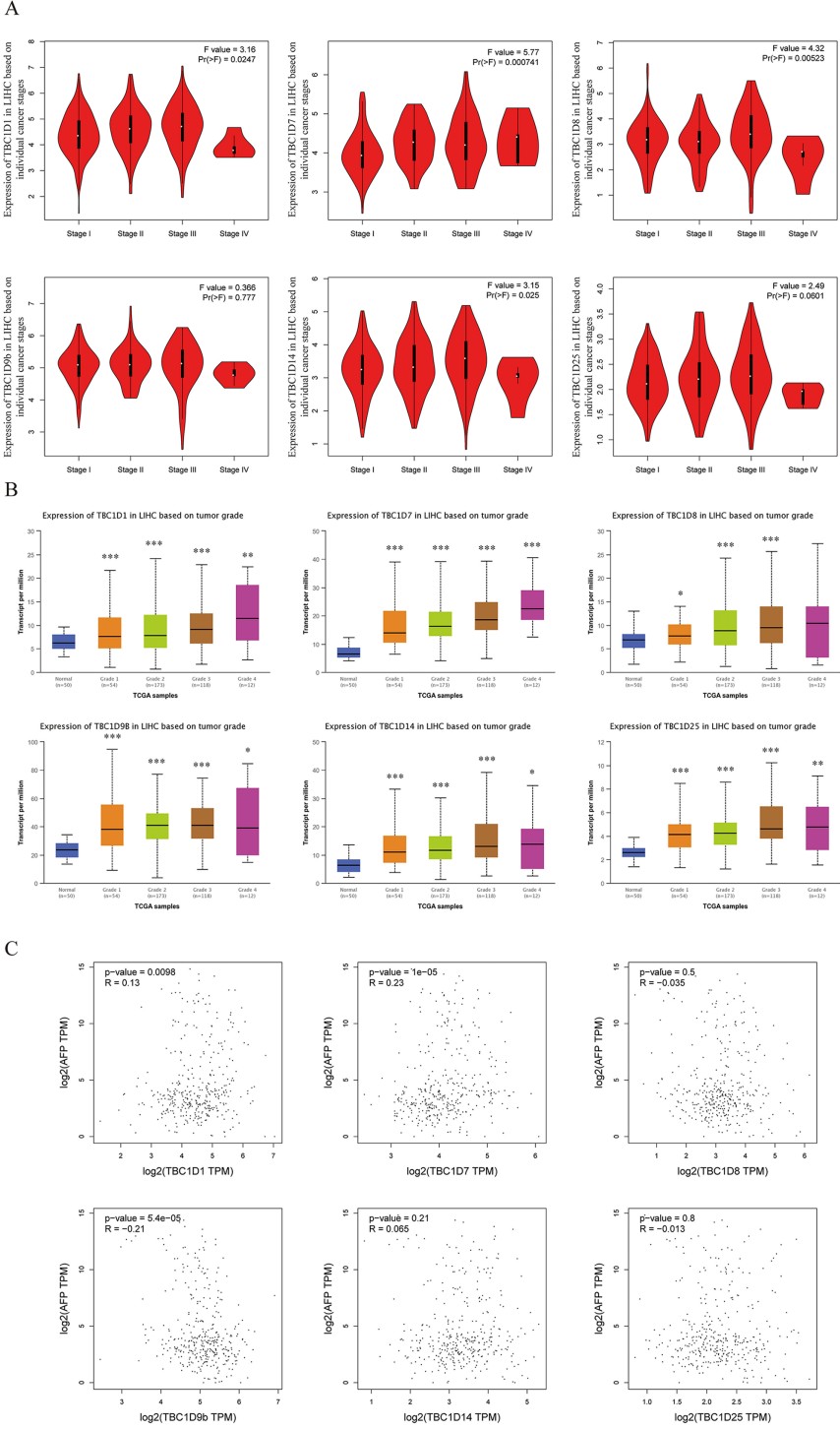

**Figure 3** **Correlation between TBC1Ds expression and HCC clinical features.** (A) The mRNA levels of TBC1Ds in healthy individuals and in patients with stages 1, 2, 3, or 4 HCC (GEPIA). (B) The mRNA expressions of TBC1Ds in healthy individuals and in subgroups of patients with HCC, stratified based on tumor grades (UALCN). (C) The correlation between the mRNA levels of TBC1Ds and AFP were performed in GEPIA database. (*P < 0.05, **P < 0.01, ***P < 0.001).

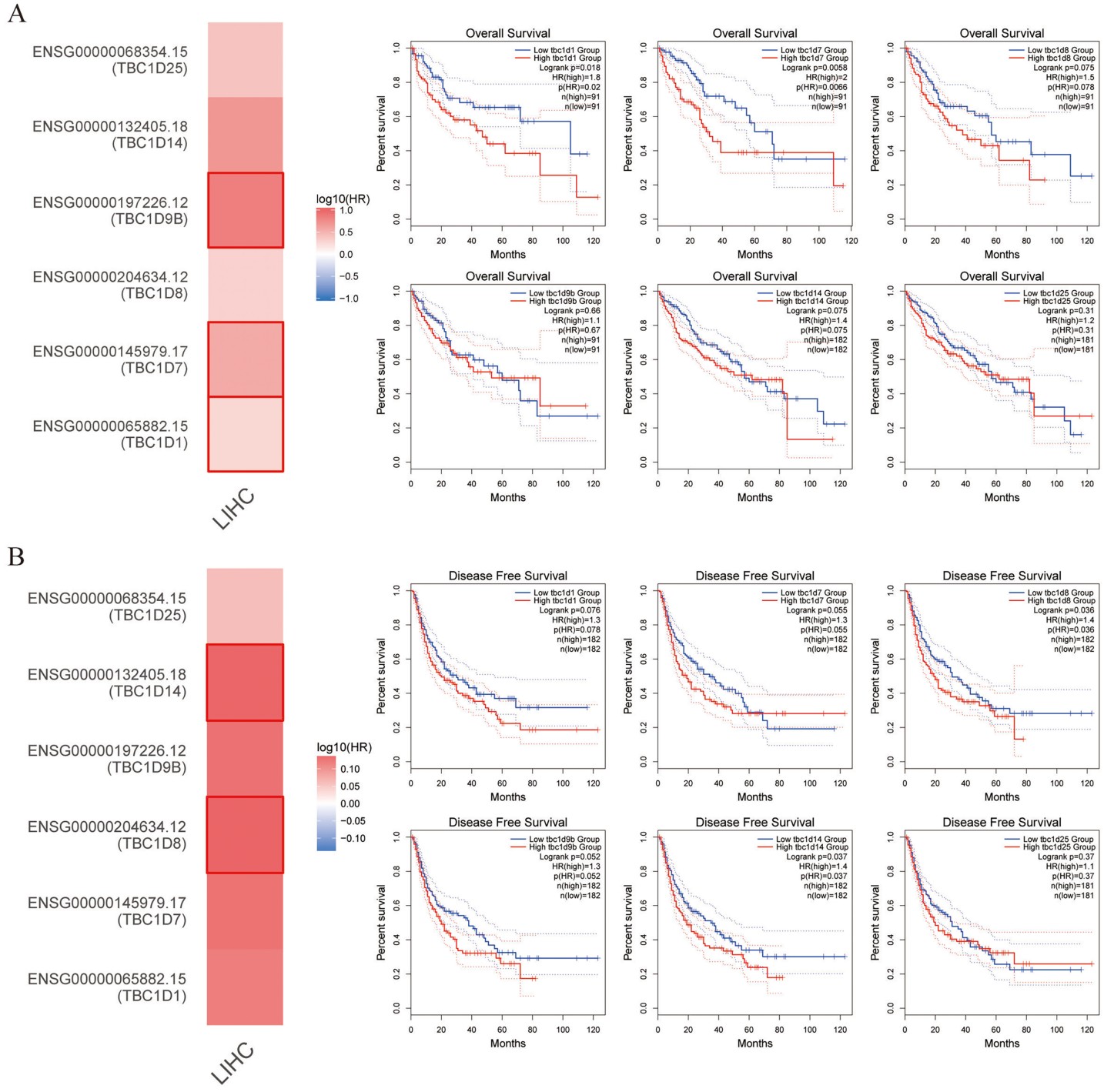

**Figure 4 Survival cure analysis of the prognostic significance of TBC1Ds expressions based on the GEPIA database.** (A) Patients with higher TBC1D1, TBC1D7 and TBC1D9b expressions showed worse OS in HCC cohorts. (B) Patients with higher TBC1D8 and TBC1D14 expressions showed worse DFS in HCC cohorts.

and the Gain was the most common alteration of TBC1D7 and TBC1D9b (Fig. 5C). Moreover, the frequency distribution of TBC1D CNV in HCC patients at different cancer stages suggested that TBC1D CNV alteration was an early event with a high frequency of

occurrence in HCC (Fig. 5D). The methylation levels of TBC1D promotors were then examined because the hypermethylation of promoters may contribute to upregulated mRNA levels of oncogenes. The results showed that only the promoter methylation level of TBC1D1 was significantly upregulated in HCC tissues (Fig. S5).

## Functions and pathways of TBC1Ds and interacting genes in HCC

To further investigate the underlying molecular mechanism of the TBC1D genes in HCC progression, the STRING tool was used to construct the network for TBC1Ds and the 48 most frequently interacting neighbor genes. The results showed that autophagy-related genes, including TSC1, TSC2, ATG3 and ATG7, were predicted to closely interact with TBC1Ds (Fig. 6A). The functions and networks of TBC1Ds and significantly associated genes were predicted using Gene Ontology (GO) and the Kyoto Encyclopedia of Genes and Genomes (KEGG) in the Metascape database. A gene set enrichment analysis (GESA) predicted the functional roles of the genes based on their molecular functions and biological processes. The results showed that GO:0051020 (GTpase binding), GO:0019901 (protein kinase binding), GO:0008047 (enzyme activator activity), GO:0010506 (regulation of autophagy), GO:0006914 (autophagy), GO:0031669 (cellular response to nutrient levels) and GO:0032006 (regulation of TOR signaling) were significantly regulated by the alterations of TBC1Ds in HCC (Figs. 6B and 6C). A KEGG analysis defined the pathways related to the functions of TBC1D alterations and the frequently altered neighbor genes. A total of seven pathways related to the functions of TBC1Ds in HCC were found through the KEGG analysis (Fig. 6D). Among these pathways, has04140: (Autophagy-animal), hsa04152: (AMPK signaling pathway), hsa04150: (mTOR signaling pathway) and hsa04137: (mitophagy-animal) have been thoroughly studied in the tumorigenesis and pathogenesis of HCC (Chao et al., 2022; Bi et al., 2021).

## Relevance between TBC1D expression and immune infiltration in HCC

The tumor microenvironment (TME) has been well established as an important mechanism influencing tumor initiation, progression and metastasis. Tumor-infiltrating immune cells (TIICs) are a significant component of the TME. To determine the effects of TBC1Ds on TIICs, the TIMER2.0 database was used to analyze the relationship between TBC1Ds and immune cell markers in HCC. A series of analyses showed that TBC1Ds, including TBC1D1, TBC1D7, TBC1D8, TBC1D9b, TBC1D14 and TBC1D25, were significantly correlated with the immune infiltration of B cells, CD4+ T cells, macrophages, neutrophils and dendritic cells. However, only TBC1D1, TBC1D8 and TBC1D25 were significantly correlated with the immune infiltration of CD8+ T cells (Fig. 7). Copy number alterations of TBC1D7 and TBC1D25, which may include arm-level gain and high amplification, were significantly correlated with the infiltrating levels of B cells, CD4+ T cells, CD8+ T cells, macrophages, neutrophils and dendritic cells. High amplification of TBC1D14 also showed a significant correlation with the infiltrating levels of macrophages (Fig. 8). To further explore the role of TBC1Ds in the tumor immune microenvironment of HCC, the correlation between the mRNA levels of TBC1Ds and the immune checkpoint receptors of PDL1 and CTLA4 was examined. The results showed that the mRNA levels of

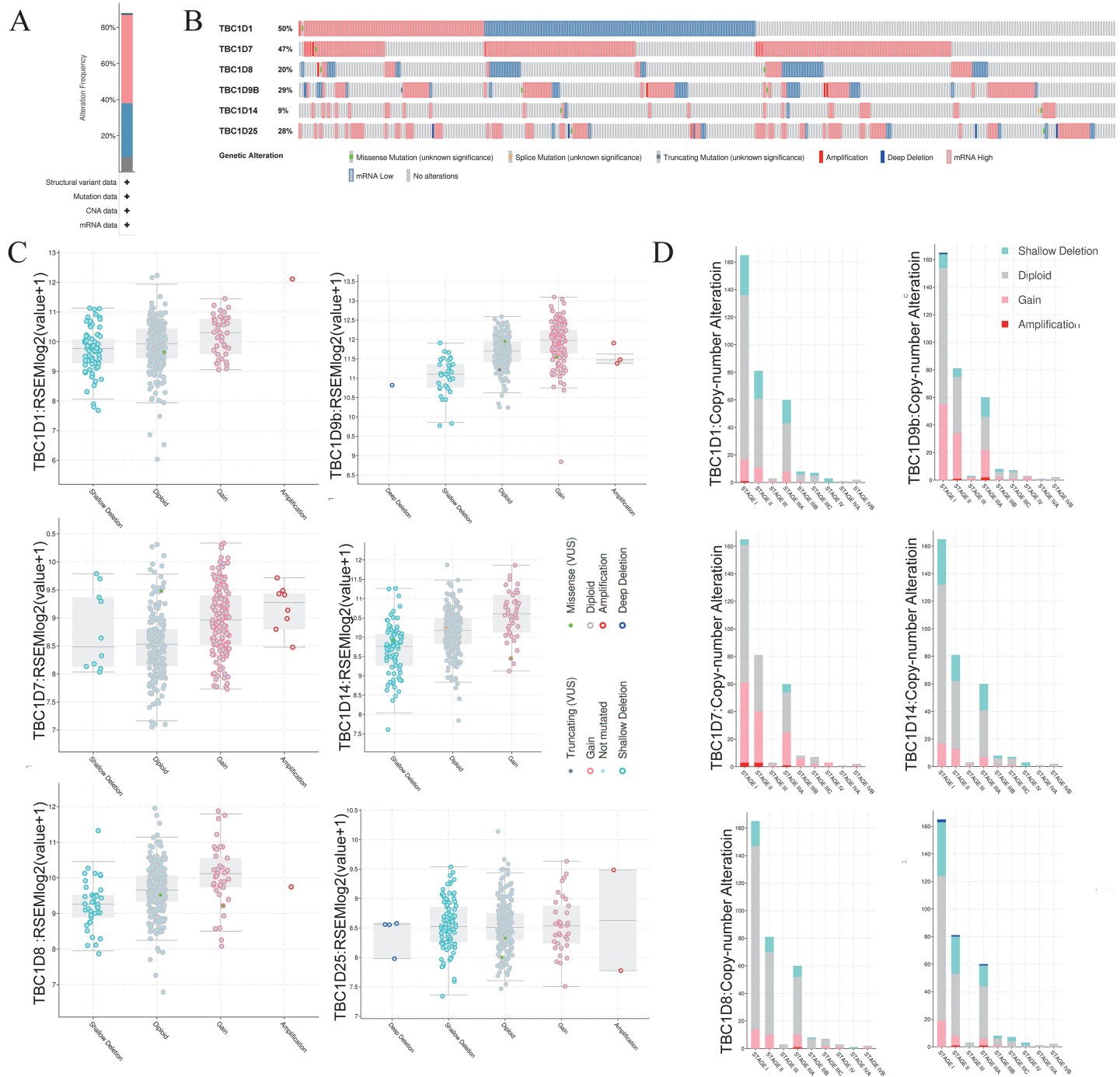

**Figure 5  The gene alterations of TBC1Ds in HCC as identified in cBioPortal.** (A) The overall gene expressions and mutation analysis of TBC1Ds in HCC. (B) OncoPrint provides an overview of the genomic alterations of each gene of TBC1Ds in HCC based on the TGCA database. (C) TBC1Ds expressions in different TBC1Ds CNV groups. (D) Distributions of TBC1Ds CNV frequency in different stage and grade and subgroups.

TBC1D1, TBC1D14 and TBC1D25 were significantly correlated with PDL1 expression and the mRNA levels of TBC1D1, TBC1D8, TBC1D14 and TBC1D25 were significantly correlated with CTLA4 expression (Fig. S6). All these results indicated that TBC1Ds may

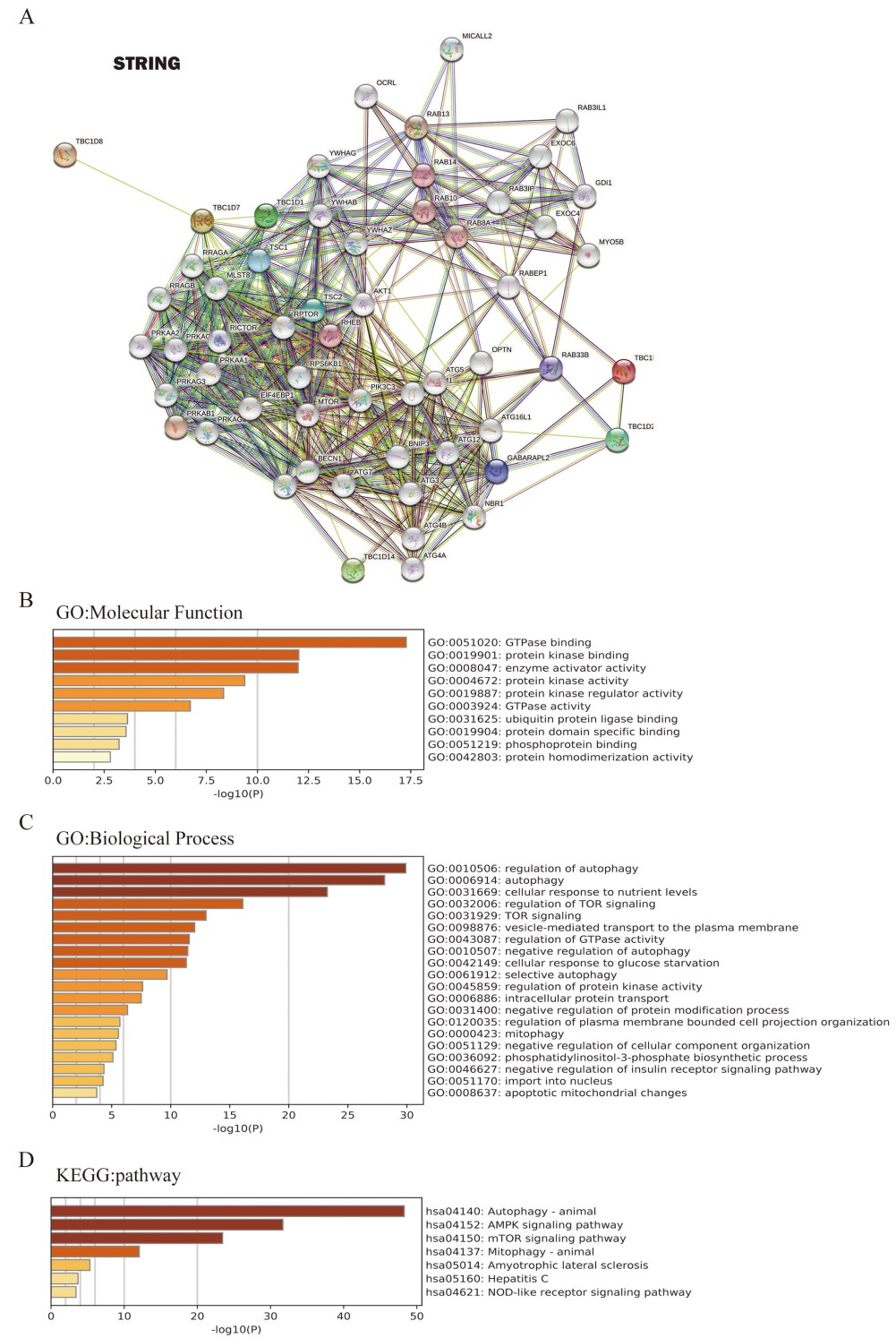

**Figure 6 TBC1Ds and related genes enrichment analysis.** (A) Predicted protein-interacted network of TBC1Ds and associated genes using the STRING tool. (B–D) The functions of and TBC1Ds and genes significantly associated with TBC1Ds were predicted by the analysis of Gene Ontology (GO), including molecular function and biological process, and Kyoto Encyclopedia of Genes and Genomes (KEGG) pathway.

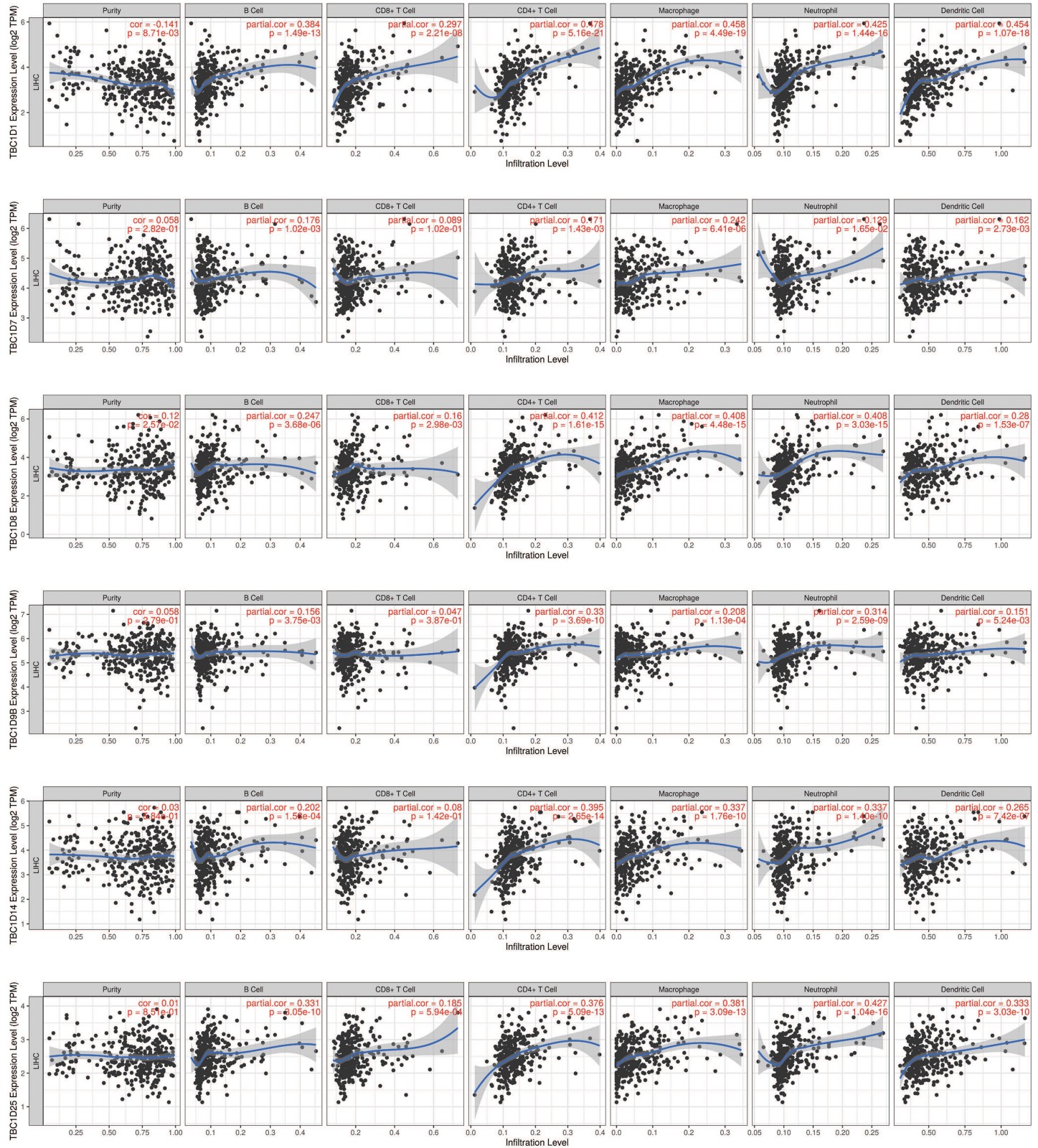

**Figure 7 The correlations between transcript levels of TBC1Ds and the immune infiltration in HCC.** Significant correlations between TBC1Ds mRNA expressions and infiltrating B cells, CD4+ T cells, CD8+ T cells, macrophages, neutrophils and dendritic cells in Hcc.

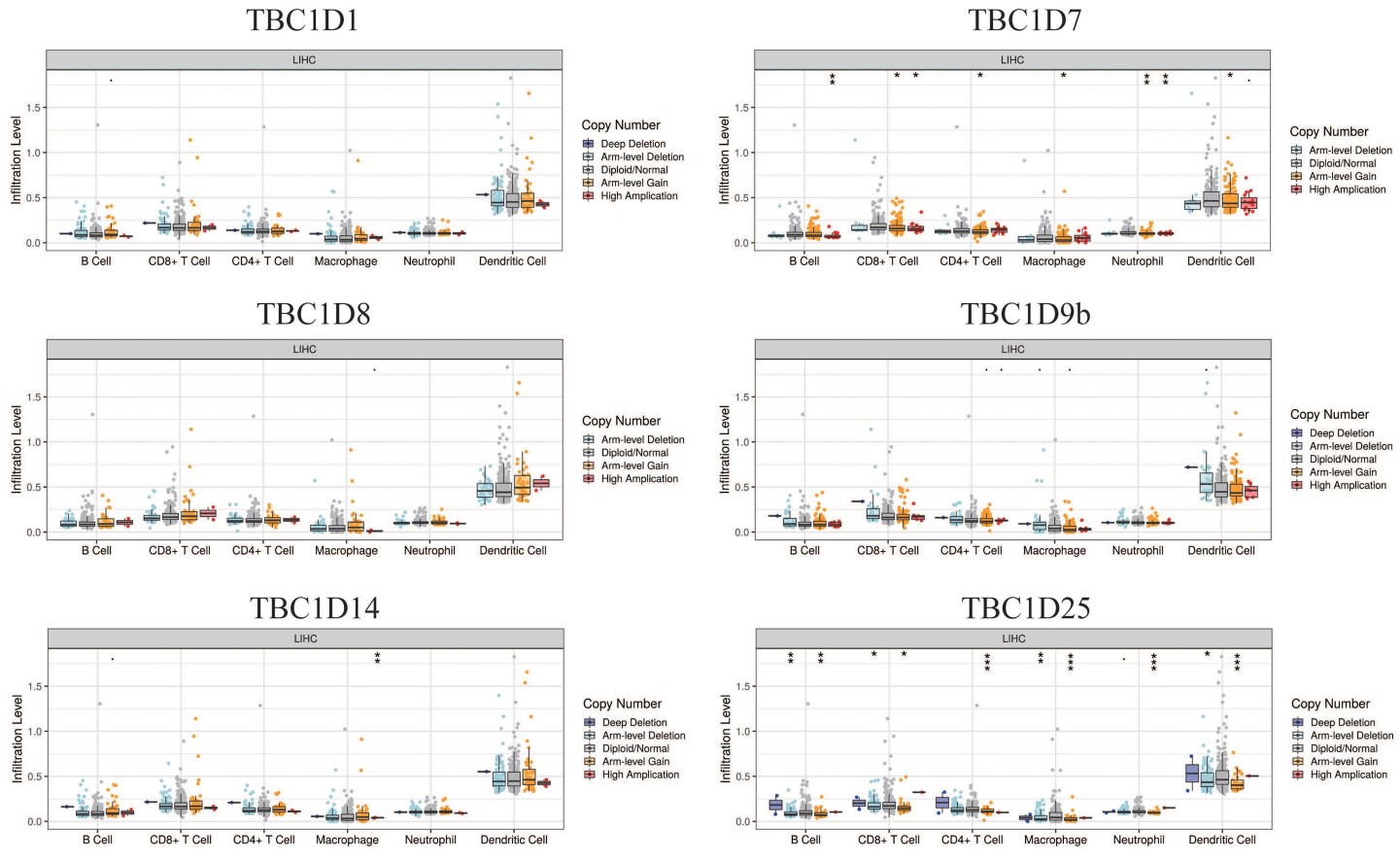

**Figure 8  The effect of TBC1D CNVs on the infiltrating levels of B cells, CD4+ T cells, CD8+ T cells, macrophages, neutrophils and dendritic cells in HCC.** (*$P < 0.05$, **$P < 0.01$, ***$P < 0.001$).

participate widely in HCC immune cell infiltration and play a vital role in the occurrence and progression of HCC.

## Knockout of TBC1D8 and TBC1D14 significantly suppressed the malignant feature of HCC cells *in vitro*

The above results indicated that TBC1Ds may be an independent factor for predicting the prognosis of patients with HCC. To further investigate the role of TBC1Ds on the malignant feature of HCC, we knockout the expression of TBC1D8 and TBC1D14 in Hep3B cells as these two genes play vital function in other types of cancers (*Chen et al., 2019*; *Lu et al., 2022*) and showed relative tighter relationship to the prognosis of patients with HCC. In the present study, Western blot analysis examined knockout efficiency (Fig. 9A) and CCK8 results showed that knockout of TBC1D8 and TBC1D14 both significantly suppressed the proliferation rate of Hep3B cells (Fig. 9B). Wound healing and EdU assays also revealed that knockout of TBC1D8 and TBC1D14 both significantly downregulated the proliferation and migration abilities of HCC cells (Figs. 9C and 9D). These experiments further verified the fact that TBC1Ds promote HCC progression *in vitro*.

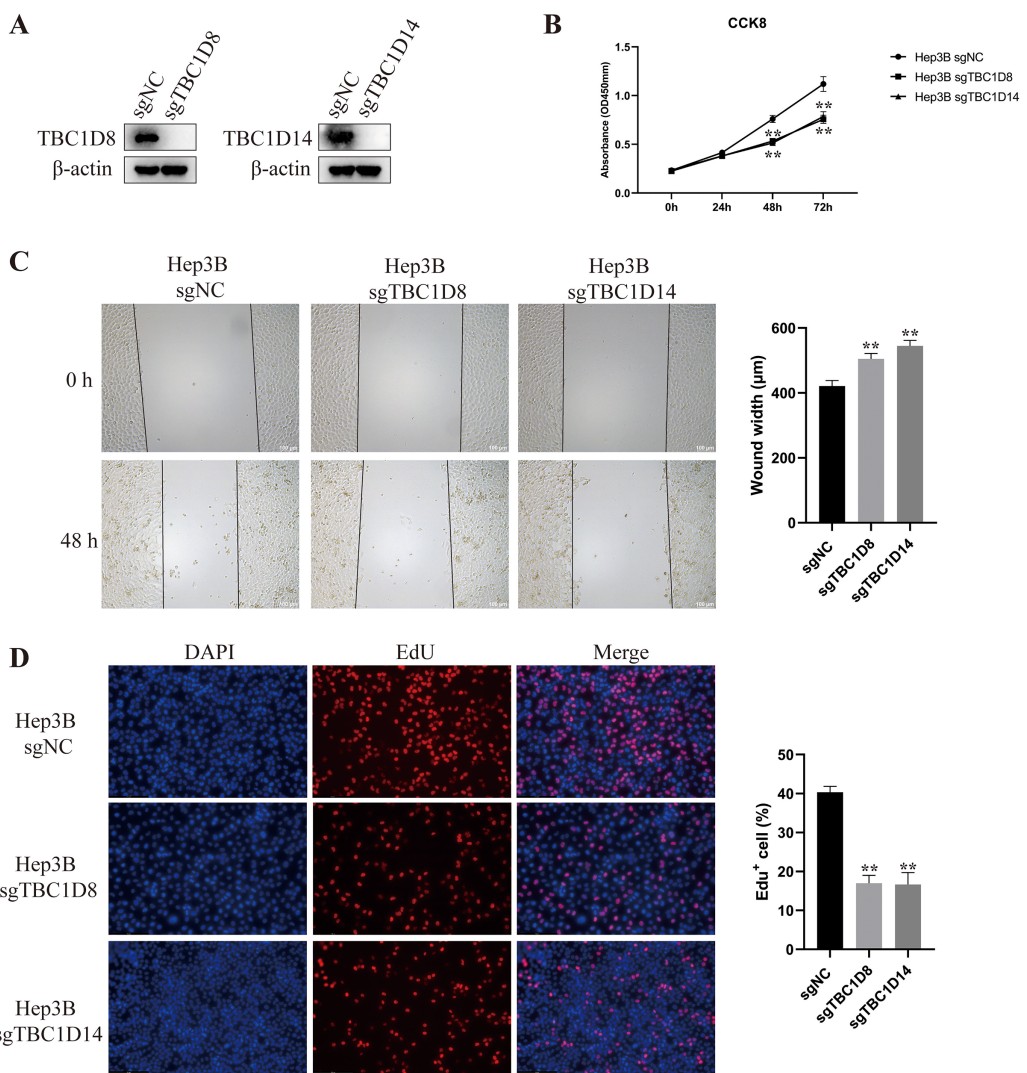

**Figure 9 Knockout of TBC1D8 and TBC1D14 suppressed the proliferation and metastasis of HCC cells *in vitro*.** (A) Western blotting analysis of protein levels of TBC1D8, TBC1D14 and β-actin upon knockout of TBC1D8 or TBC1D14 in Hep3B cells. (B) CCK8 assay of Hep3B cells after knockout of TBC1D8 or TBC1D14 compared to controls. (C and D) Representative images of the wound healing and EdU assays that were performed with Hep3B cells after knockout of TBC1D8 or TBC1D14 compared to controls. The width of the wound and the percentage of EdU positive cells were calculated and are shown in the right bar graph. ($^*P < 0.05$, $^{**}P < 0.01$).

## DISCUSSION

The progression of HCC is swift and violent and the 5-year survival rate of patients with HCC is below 10%, so research on the pathogenesis of HCC and potential therapies for HCC is needed (*Villanueva, 2019*). Although dysregulation of TBC1D family members has been studied in many cancers, the roles that TBC1Ds played in the process and prognosis of HCC have not been confirmed. This study is the first to explore the mRNA expression and prognostic values of different TBC1Ds in HCC. The findings of this study add to the knowledge of the pathogenesis of HCC and prognosis of HCC patients.

Many previous studies have shown that the AMPK-TBC1D1 signal nexus regulates glucose uptake and TBC1D1 modulates the expression of GLUT1 and GLUT4 on the cell surface (*Kjobsted et al., 2019*; *Henriques et al., 2020*; *Benninghoff et al., 2020*). Although there are not many studies on TBC1D1 in cancer, the present study identified the potential of TBC1D1 as a biomarker of HCC. The TIMER and GEPIA datasets revealed that the expression of TBC1D1 was higher in HCC than in normal tissues, and the UALCN dataset showed that TBC1D1 expression was also correlated with the clinical characteristics of patients with HCC. The Human Protein Atlas Database was then used to verify that the protein level of TBC1D1 was higher in HCC than in normal tissues. High TBC1D1 expression was significantly associated with poor OS in all of the patients with HCC. Previous studies have demonstrated the relationship between tumor occurrence and the uncontrolled immune regulation of tumors (*Chen & Mellman, 2017*). In this study, the analysis of the TIME database showed that TBC1D1 level was significantly associated with the infiltrating levels of TIICs and tumor immune invasion.

TBC1D7 has been shown to be a vital part of the TSC1-TSC2 complex and dysregulation of TBC1D7 leads to delayed induction of autophagy (*Dibble et al., 2012*). TBC1D7 has also been shown to promote melanoma cell invasion and melanoma metastasis (*Qi et al., 2020*). In the present study, the expression of TBC1D7 in HCC tissues was higher than in normal tissues, and TBC1D7 expression was significantly correlated with cancer stage and tumor grade. High TBC1D7 expression was significantly correlated with poor OS in HCC patients. TBC1D7 expression was also significantly correlated with the infiltrating levels of B cells, CD4+ T cells, macrophages, neutrophils and dendritic cells in HCC.

The function of TBC1D8 has been studied in ovarian cancer, where it was found to be significantly upregulated. By binding to PKM2, TBC1D8 hinders PKM2 tetramerization to decrease pyruvate kinase activity and promote aerobic glycolysis, leading to ovarian cancer tumorigenesis and metabolic reprogramming (*Chen et al., 2019*). The present study found that the expression of TBC1D8 was higher in HCC tissues than in normal liver tissues and its expression was significantly correlated with cancer stage and tumor grade. High TBC1D8 expression was significantly correlated with poor DFS in all of the patients with HCC. The expression of TBC1D8 was also significantly correlated with the infiltrating levels of B cells, CD4+ T cells, CD8+ T cells, macrophages, neutrophils and dendritic cells in HCC.

The functions of TBC1D9b, TBC1D14 and TBC1D25 in cancer have not been well studied but are known to be closely related to autophagy. TBC1D9b has been found to interact with LC3B and other mammalian ATG8 homologues, and the interacting domain of TBC1D9b with LC3 is unique and different from previously known LC3-interacting regions defined in other interactions. TBC1D9b could inactivate RAB11A through GTPase activity and facilitate proper autophagic flux (*Liao et al., 2018*). TBC1D14 localizes at the Golgi complex during amino acid starvation and interacts with ULK1 and RAB11. When overexpressed, TBC1D14 causes the tubulation of ULK1-positive endosomes and inhibits autophagy (*Longatti et al., 2012*). TBC1D25 interacts with the ATG8 family proteins *via* a LIR/LRS-like sequence and regulates the interaction of the autophagosome with

lysosomes. When overexpressed, TBC1D25 inhibits the maturation of the autophagosome (*Itoh et al., 2008*; *Hirano et al., 2016*). In the present study, TBC1D9b, TBC1D14 and TBC1D25 were significantly overexpressed in HCC, and the expression levels of TBC1D9b, TBC1D14 and TBC1D25 were significantly correlated with HCC tumor grade. Moreover, the expression level of TBC1D14 was significantly correlated with HCC cancer stage. High TBC1D9b expression was significantly correlated with poor OS in HCC patients, and high TBC1D14 expression was significantly correlated with poor DFS in all of the patients with HCC. The TIMER database showed that the expressions of TBC1D9b, TBC1D14 and TBC1D25 were significantly correlated with the immune infiltration of B cells, CD4+ T cells, macrophages, neutrophils and dendritic cells, and the copy number alteration of TBC1D25 was significantly correlated with the infiltrating levels of B cells, CD4+ T cells, CD8+ T cells, macrophages, neutrophils and dendritic cells.

This study used the STRING and Metascape databases to analyze the predicted network of TBC1Ds and frequently interacting neighbor genes. GSEA and KEGG analysis showed that TBC1Ds and their interacting genes were mainly responsible for the regulation of autophagy, AMPK and the mTOR signaling pathway. AFP is a vital gene signature for the clinical diagnosis of HCC and the levels of serum AFP are closely correlated with the progressive grade of HCC (*Heimbach et al., 2018*; *Wong, Ahmed & Gish, 2015*; *Li et al., 2009*). Recent studies have shown that AFP can interact with a list of proteins to promote the progression of HCC (*Li et al., 2009*; *Zhang et al., 2015*) and can inhibit autophagy by regulating the mTOR pathway in HCC cells (*Wang et al., 2018*). Autophagy can degrade AFP aggregates in HCC cells (*Zhao et al., 2017*), and previous studies have shown that TBC1Ds participate in the process of autophagy, so it is worth analyzing the correlation between TBC1Ds and AFP in human HCC. In this study, the mRNA levels of TBC1D1, TBC1D7 and TBC1D9b had strong positive correlations with the mRNA level of AFP. Based on the results of the present study, the correlation between serum level of AFP and TBC1D protein levels in HCC tissues will be explored in future studies.

To further verify the promoting role of TBC1Ds in HCC progression, two TBC1D family members were knocked out in HCC cells. Previous studies have revealed the roles of TBC1D8 and TBC1D14 in other types of cancers, so these two TBC1Ds were knocked out in Hep3B cells in this study. The protein expressions of TBC1D8 and TBC1D14 were significantly upregulated and significantly correlated with the prognosis of patients with HCC. A series of *in vitro* experiments demonstrated that knockout of TBC1D8 or TBC1D14 significantly inhibited the proliferation and migration abilities of HCC cells.

This study systematically analyzed the expression and prognostic value of TBC1Ds in HCC and further verified the promoting role of TBC1D8 and TBC1D14 in HCC progression *in vitro*. The results of this study suggest that TBC1Ds may be promising biomarkers for the early diagnosis of HCC.

## CONCLUSION

The results of this study revealed that TBC1D1, TBC1D7, TBC1D8, TBC1D9b, TBC1D14 and TBC1D25 are overexpressed in HCC and closely associated with the prognosis of patients with HCC. A gene enrichment analysis of interacting genes showed that TBC1Ds

play a vital role in regulating autophagy in HCC. A series of *in vitro* studies further demonstrated that TBC1D8 and TBC1D14 promote the proliferation and metastasis of HCC cells. All these results suggest that TBC1Ds may serve as new therapy targets for HCC.

### Funding

This work was supported by the Research Foundation of jiangsu provincial commission of health and family planning (BJ15024), the Science and Technology Development Fund of Nanjing Medical University (NMUB20210170), and the Jiangsu Provincial Elderly Health Research Project (No. LK2021011). The funders had no role in study design, data collection and analysis, decision to publish, or preparation of the manuscript.

### Grant Disclosures

The following grant information was disclosed by the authors:
Research Foundation of Jiangsu Provincial Commission of Health and Family Planning: BJ15024.
Science and Technology Development Fund of Nanjing Medical University: NMUB20210170.
Jiangsu Provincial Elderly Health Research Project: LK2021011.

### Competing Interests

The authors declare that they have no competing interests.

### Author Contributions

- Pei Zhang performed the experiments, prepared figures and/or tables, and approved the final draft.
- Lei Zhu analyzed the data, prepared figures and/or tables, and approved the final draft.
- Xiaodong Pan conceived and designed the experiments, authored or reviewed drafts of the article, and approved the final draft.

### Data Availability

The raw measurements are available in the Supplemental File.
The third-party raw data are available at:
- TIMER, https://cistrome.shinyapps.io/timer
- UALCAN, http://ualcan.path.uab.edu
- GEPIA, http://gepia.cancer-pku.cn
- cBioPortal, http://www.cbioportal.org
- STRING, https://cn.string-db.org
- The Metascape database, https://metascape.org/gp/index.html#/main/step1
- The Human Protein Atlas, https://www.proteinatlas.org.

## Supplemental Information

Supplemental information for this article can be found online at http://dx.doi.org/10.7717/peerj.17362#supplemental-information.

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
