# Peer review of "A comprehensive analysis of the oncogenic and prognostic role of TBC1Ds in human hepatocellular carcinoma"

_PeerJ, doi:10.7717/peerj.17362_

## Round 0.1 · original submission · Major Revisions

The manuscript is well-written overall, but some grammatical errors need correction. Additionally, your analysis of TBC1Ds expression levels should be expanded to include data from the GEO database, not just the TCGA database.

Improvement is needed in the presentation and analysis of your figures. This includes the need for statistical analysis of IHC values, discussion on the clinical significance of TBC1Ds and AFP, and expanding the prognosis analysis to include recurrence-free survival.

Please clarify your rationale for focusing on TBC1D8 and TBC1D14 in your knockout experiments.

Please discuss the role of TBC1Ds in other types of liver cancer, such as intrahepatic cholangiocarcinoma.

Reviewer 1 ·

Basic reporting

Dear editor, overall, the manuscript writing approach is good. But there are many major mistakes which have compromised the quality of the work. Most of the mistakes are highlighted in the main manuscript and some comments are added too. The authors need to rephrase most of the sentences to remove the grammatical and inappropriate uses of words. For example, In the results section instead of focusing on their own results they have discussed the results of previous studies, which really make the paper unacceptable at the current format.

Experimental design

Overall, the experimental approach is good but need to focus on rewriting to make it more clear.

Validity of the findings

The findings are confusing, the authors need to make it more clear in the results section, that why they only select TBC1D8 and TBC1D14 for knock out experiments.

Additional comments

Dear Editor, the manuscript aim and objectives is interesting but the author failed to explain in a professional way. The authors need to rewrite the whole manuscript to make it clear all the doubts, mostly highlighted in the manuscript.

Reviewer 2 ·

Basic reporting

The study explored the diverse expression patterns and prognostic values of TBC1Ds in hepatocellular carcinoma. They showed that TBC1Ds expression are upregulated in HCC and negatively correlated with prognosis.
The article carries on innovative analysis of the underlying the occurrence and process of HCC.

There are some concerns that need to be addressed. Below are my specific comments:

1. The manuscript was well written in general, except for some grammatical mistakes, for example in Results 3: “TBC1D1, TBC1D7, TBC1D8 and TBC1D14 groups significantly varied, whereas TBC1D9b and TBC1D25 did not significantly differ”.

2. In the Figure 1 and Figure 3, the authors only examine the mRNA levels of TBC1Ds in TCGA database. The mRNA levels of TBC1Ds in HCC should also be examined in GEO database.

3. The authors only present the graph of IHC staining of TBC1Ds in HCC, they maybe want to make the statistics analysis of IHC value of TBC1Ds in HCC sections.

4. The clinical significance of association between TBC1Ds and AFP deserve more in discussion part.

5. In Figure 4, the authors only checked the OS and PFS of patients with HCC. The RFS of patients with HCC should also be discussed for the prognosis part.

6. In Figure 7, the authors investigated the correlation between the mRNA levels of TBC1Ds and the immune cell infiltration level. The authors could also explore the correlation between TBC1Ds and the Immune checkpoint receptors expression in HCC.

7. In Figure 8, the authors knockout TBC1D8 and TBC1D14 in HCC cells. The authors maybe want to present more reasons for choosing TBC1D8 and TBC1D14 for in vitro settings.

8. As hepatocellular carcinoma is one single type of liver cancer, other types of liver cancer, for example intrahepatic cholangiocarcinoma, need some discussion about TBC1Ds.

9. This study mainly focused on the upregulation of TBC1Ds in HCC and the downstream function of TBC1Ds. As the hypermethylation of promoters contribute to the upregulated mRNA levels of oncogenes, authors maybe want to explore the methylation levels of TBC1Ds in HCC.

Experimental design

The related comments have been provdied in the Basic reporting.

Validity of the findings

The related comments have been provdied in the Basic reporting.

Additional comments

I have no additional comments.

---

## Round 0.2 · accepted · Accept

This study provides an in-depth exploration of the potential role of TBC1D family members in the progression of hepatocellular carcinoma (HCC), with detailed content and insightful analysis. By employing various online databases, the authors comprehensively compared the mRNA and protein expression levels of TBC1Ds in HCC and peritumor tissues. Furthermore, this study investigated the possible functional pathways through which TBC1Ds influence HCC. Finally, the authors further verified their hypothesis through in vitro experiments, demonstrating the suppressive role of TBC1Ds in the proliferation and migration of HCC cells.

Overall, this research offers a novel perspective and draws meaningful conclusions. The authors' use of a series of cell function assays to support their arguments enhances the credibility of the research findings. The study is well-designed, rigorously analyzed, and thoroughly argued, making it a valuable contribution to our understanding of the role of the TBC1D family in the development and progression of liver cancer. This article is worthy of reference and inspiration for researchers in the field of hepatocellular carcinoma.

Reviewer 1 ·

Basic reporting

Improved

Experimental design

Clear and Improved

Validity of the findings

Improved

Reviewer 2 ·

Basic reporting

This is an interesting study and increased our understanding of TBC1D family members influencing HCC progression.

Experimental design

The overall logic is clear and the exploration covers multiple aspects of HCC. The authors explored the expression, prognostic role, genomic alterations of SEC24C in HCC. Moreover, they also conducted the function enrichment, immune infiltration of TBC1Ds in HCC.

Validity of the findings

Tthey knockout the expression of TBC1Ds in HCC cell lines and estimated the proliferation and migration abilities of TBC1Ds in HCC. The whole experiment is very complete and convincing.

Additional comments

None.